# Phylogeographic History of Tomato Chlorosis Virus

**DOI:** 10.3390/v17040457

**Published:** 2025-03-22

**Authors:** Kangcheng Wu, Shiwei Zhang, Wende Huang, Zhenguo Du, Fangluan Gao, Xiayu Guan

**Affiliations:** 1Institute of Plant Virology, Fujian Agriculture and Forestry University, Fuzhou 350002, China; wukangch@126.com (K.W.); swayzhang@fafu.edu.cn (S.Z.); duzhenguo1228@163.com (Z.D.); 2School of Medical Technology and Engineering, Fujian Health College, Fuzhou 350101, China; 3Key Laboratory of Ministry of Education for Genetics, Breeding and Multiple Utilization of Crops, College of Horticulture, Fujian Agriculture and Forestry University, Fuzhou 350002, China; 3225216027@stu.fafu.edu.cn

**Keywords:** tomato chlorosis virus, Bayesian phylodynamics, virus dispersal

## Abstract

Tomato chlorosis virus (ToCV), first reported in Florida, USA, in 1998, has since emerged in multiple regions worldwide, posing a significant threat to global tomato production. However, its origin, migration patterns, and evolutionary history remain poorly understood. In this study, we used Bayesian phylogeographic analysis of coat protein gene sequences from 155 ToCV isolates to reconstruct its phylogeographic history. Our results show that ToCV evolves at a rate of 6.24 × 10^−4^ subs/site/year (95% credibility interval: 4.35 × 10^−4^–8.28 × 10^−4^), with the most recent common ancestor dating back to 1882. The maximum clade credibility (MCC) tree revealed three major clades, with Clade 1—whose most recent common ancestor dates to approximately 1975—comprising over 90% of the isolates. Although the exact origin of ToCV remains uncertain, we identified five distinct migration pathways: one from Europe to the Americas, one from Europe to South Asia, one from the Middle East to East Asia, one from East Asia to mainland China, and one from mainland China to Europe. These findings underscore the complex global spread of ToCV and suggest that multiple geographic areas have contributed to its ongoing evolution and dissemination.

## 1. Introduction

Tomato chlorosis virus (ToCV), a member of the genus *Crinivirus* within the family *Closteroviridae* [1], is a highly destructive pathogen that poses a serious threat to tomato production worldwide [2]. With a remarkably broad host range, it is capable of infecting over 119 plant species spanning 28 families, including members of economically important groups like *Aizoaceae*, *Brassicaceae*, and *Solanaceae* [3]. Within these, tomatoes and peppers from the *Solanaceae* family are particularly susceptible. Early symptoms of ToCV in tomato plants are characterized by chlorotic, polygonal areas limited by the main veins, progressing into interveinal bright yellowing that starts on lower leaves and spreads upwards. As the disease advances, the leaves display reddish-brown necrotic flecks, rolling, thickening, and a crispy texture, leading to weakened plant growth, reduced fruit development, and delayed ripening without noticeable symptoms on the fruit itself [2]. In agricultural settings, ToCV often occurs in conjunction with other major viruses, such as tomato yellow leaf curl virus (TYLCV), tomato severe rugose virus (ToSRV), and pepino mosaic virus (PepMV), creating compounded infections that exacerbate crop damage and amplify economic losses [4,5,6].

Like other members of the *Crinivirus* genus, ToCV possesses a bipartite genome composed of two positive-sense single-stranded RNAs (RNA1 and RNA2) [1]. RNA1 contains four open reading frames (ORFs), which are primarily dedicated to synthesizing proteins necessary for viral replication. Conversely, RNA2 contains nine ORFs that encode proteins essential for assembling the viral coat, facilitating cell-to-cell movement and vector-mediated transmission. ToCV has two coat proteins, known as the major coat protein (CP) and the minor coat protein (CPm). Of these, the CP plays a crucial role in virion assembly and the establishment of viral infectivity within the host. Despite its relatively short length, CP has consistently proven to be an indispensable target for evolutionary studies across a wide range of RNA viruses, likely due to the fact that CP typically has the highest number of available sequences for most viruses [7,8,9,10,11,12].

ToCV is transmitted in a semi-persistent manner under natural conditions by whiteflies of the genera *Bemisia* and *Trialeurodes*, facilitating its broad distribution [3,13]. Since its initial identification in Florida, USA, in 1998, the virus has been documented in over 30 countries across multiple continents [2]. The ongoing geographic spread of ToCV raises increasing concerns due to its potential to affect agricultural systems on a global scale.

Understanding the evolution and spread of emerging plant viruses is critical for developing effective and sustainable management strategies for plant diseases, ultimately benefiting both researchers and farmers in combating crop-related challenges. Despite its significance, the evolutionary dynamics of ToCV remain underexplored. To gain a deeper insight into the global phylogeographic history of ToCV, we conducted a phylodynamic analysis to investigate its spatial dispersal dynamics. Nucleotide sequences from the CP gene of ToCV, retrieved from GenBank, were utilized in this study. These sequences represent viral isolates from 15 countries over a 26-year period, with a subset sequenced in our own lab.

## 2. Materials and Methods

Coat protein gene sequences of 155 ToCV isolates with known sampling dates and geographic locations were retrieved from GenBank (access on 15 October 2024). These isolates were collected from 15 countries between 1997 and 2023. Detailed information about these isolates, including their host origin and collection date, is provided in Appendix A. To reduce computational complexity and increase post-analysis interpretability, we grouped the isolates into six distinct geographic regions: Mainland China (MC, *n* = 67), East Asia (EA, *n* = 23), South Asia (SA, *n* = 8), Europe (EU, *n* = 19), the Middle East (ME, *n* = 27), and America (AM, *n* = 11, Figure 1a). Notably, a region may be identified as the source of the virus due to over-representation within the dataset. Among these regions, mainland China, which accounts for the largest sample size in this study, may be erroneously inferred as the virus’s origin during subsequent phylogeographic analysis. To address this potential sampling bias, we assembled 10 subsampled datasets by randomly selecting 30 sequences from mainland China to compare with results from the original dataset. This subsampling approach ensures a sufficient sample size for the reliable estimation of population genetic parameters.

Multiple sequence alignments were performed using the MAFFT codon-based algorithm [14] implemented in PhyloSuite 1.23 [15]. Prior research has highlighted the risk of inaccurate evolutionary rate estimates that arise when recombinants are included in evolutionary analyses. This issue is further compounded by the challenges of accounting for recombination in phylodynamic studies. To address this, we screened the aligned sequences for recombination by employing a suite of methods, including RDP, GENECONV, BOOTSCAN, MAXCHI, CHIMAERA, SISCAN, and 3SEQ, as implemented in RDP 5.64 [16]. To minimize false positives, only recombination events supported by at least four of the seven methods with *p* < 10^−6^ were considered. No significant recombination signals were detected, allowing us to proceed with the complete dataset for subsequent analyses.

To assess the presence of temporal structure in the sequence data, we conducted a regression analysis of phylogenetic root-to-tip distances against sampling date using TempEst 1.5 [17]. The analysis relied on tree topology and branch lengths using a maximum likelihood under the HKY + *Г*_4_ substitution model, chosen based on the Bayesian information criterion, as determined by ModelTest-NG [18]. Furthermore, we implemented a clustered-permutation date-randomization test (DRT) analysis to rigorously evaluate the temporal signal within the dataset [19]. However, both methods failed to detect any temporal structure within the data. Consequently, we utilized our previous estimate of the substitution rate of the CP gene to calibrate the evolutionary timescale of ToCV [20].

To investigate the spatio-temporal dynamics of ToCV, a discrete Bayesian phylogeographic model was applied using an asymmetric continuous-time Markov chain framework with Bayesian stochastic search variable selection [21], implemented in the program BEAST 1.10.6 [22]. Six geographical regions, as previously described, were encoded as discrete states. Well-supported pairwise diffusions were identified using Bayes factors in SpreaD4 [23]. The HKY + *Г*_4_ substitution model, as described above, was used, and a uniform prior of 6.08 × 10^−4^–1.73 × 10^−3^ substitutions/site/year was specified for the absolute substitution rate of the CP gene, based on our previous estimate [20]. Marginal likelihoods were calculated through path sampling and stepping-stone sampling to compare the constant-size, exponential-growth, and Bayesian skyline coalescent tree priors, as well as to evaluate clock models, including the strict and uncorrelated lognormal relaxed clock [24]. A Bayesian skyline coalescent tree prior and uncorrelated lognormal relaxed clock provided the best fit to our sequence data (Appendix A). All Markov Chain Monte Carlo (MCMC) simulations, spanning at least 500 million generations, ensured adequate convergence and robust sampling, with all parameters achieving estimated sampling sizes (ESS) exceeding 200. The first 10% of the sample were discarded as burn-in, with sampling performed every 50,000 steps. In addition, the number of expected location-state transitions, representing viral migration events between geographic regions over the course of evolutionary history, was estimated using Markov jump counts [25].

To gain deeper insights into the temporal migration pattern of ToCV, the MCC tree annotated from the previous MCMC runs was analyzed using the ETE toolkit [26] in combination with in-house scripts developed by Brynildsrud et al. [27]. In this analysis, migration events were modeled as occurring on nodes. While this modeling approach might introduce a slight bias, particularly by inflating the estimated ages of early migration events, the effect is minimal for more recent migrations due to the extensive branching of the phylogenetic tree [27].

## 3. Results

### 3.1. Temporal Dynamics of Tomato Chlorosis Virus

The maximum clade credibility (MCC) tree topology inferred from our phylogenetic analysis revealed that ToCV isolates clustered into three distinct clades (Figure 1b). Clade I exhibited substantial diversity, encompassing a wide range of geographic regions and host species. Clade II, in contrast, was more geographically restricted, including seven isolates from Europe and two from Asia (India and Republic of Korea). Clade III was the most narrowly defined, comprising just four isolates, all from China. Our Bayesian phylogenetic analyses placed the most recent common ancestor of ToCV at 1882 CE (95% credibility interval 1786–1960 CE). However, the analysis could not definitively pinpoint the geographic origin of the root node. Although Europe showed a slightly higher probability than the Middle East, the Americas, and East Asia, the value did not exceed 0.35, making the result inconclusive.

The estimated mean substitution rate for the CP gene of ToCV was 6.24 × 10^−4^ subs/site/year (95% credibility interval 4.35 × 10^−4^–8.28 × 10^−4^). This rate is slightly lower than our previous estimate of 1.12 × 10^−3^ subs/site/year (95% credibility interval 6.08 × 10^−4^–1.73 × 10^−3^) reported by Zou et al. [20].

### 3.2. Phylogeographic History of Tomato Chlorosis Virus

Our Bayesian phylogeographic analysis identified five significant migration pathways, each supported by high diffusion rates. Three of these pathways involved Europe: two emigrations, one to the Americas and the other to South Asia, and one immigration from Mainland China. The other two pathways were from East Asia to Mainland China and from the Middle East to East Asia (Figure 2a). These migration pathways were further supported by the results from 10 subsampled datasets (Appendix A). The mean diffusion rates ranged from 0.826 to 1.853 migration events per lineage per year (Figure 2b). The highest mean rate was observed for migration from East Asia to Mainland China, whereas the lowest was for migration from Europe to the Americas. The inferred spatial dynamics of ToCV suggest that Europe served as a significant hub for epidemics that spread to other regions.

In addition, our phylogeographic analysis indicated that the mean Markov rewards for East Asia (319, 95% credibility interval 179–526) were slightly higher than those for Europe (277, 95% credibility interval 102–457). This suggests that East Asia has played a dominate role in the evolution and persistence of ToCV isolates over the investigated time period.

### 3.3. Temporal Migration Pattern of Tomato Chlorosis Virus

Although ToCV may have emerged as early as 1882 CE, its migration across geographic regions did not commence until the late 1920s, with the exception of movement from Europe to East Asia (Figure 2c). Between 1920 and 1960, these migration events were generally limited, apart from a notable wave of migration from East Asia to Mainland China. From 1960 onwards, intra-regional migration became the predominant pattern of ToCV dispersion in most regions, with South Asia being a notable exception.

## 4. Discussion

In this study, we applied Bayesian phylogeographic analysis to investigate the evolutionary history and global spread of ToCV, a significant threat to tomato production. Our findings offer useful insights into the virus’s global dynamics and may help inform future disease management strategies.

The evolutionary rate of tomato chlorosis virus (ToCV) estimated in this study, approximately 6.24 × 10^−4^ subs/site/year, is in line with the rates observed in other plant viruses [9]. However, this rate is notably lower than our previous estimate of 1.12 × 10^−3^ subs/site/year [23]. This discrepancy can likely be attributed to the differing datasets used in the two studies. The current analysis involved a larger number of isolates, most of which were collected after 2014. This suggests that ToCV may have undergone a phase of population expansion prior to 2014. Viruses experiencing population expansion typically exhibit accelerated mutation rates, as a rapid increase in genetic diversity within the population can lead to a higher rate of evolution [28]. This may explain the elevated evolutionary rate observed in our earlier analysis.

With the newly estimated rate, the most recent common ancestor (MRCA) of ToCV is dated back to 1882, more than 100 years before the yellow leaf disorder syndrome associated with this virus was first observed in Florida in 1989. This implies that ToCV could have been circulating undetected for over a century before it was formally identified as a pathogen. Although this finding might seem unexpected, several factors could account for this long, unnoticed circulation. First, ToCV has a relatively broad host range, including some wild plant species, which might have allowed it to persist undetected in natural reservoirs [3]. Second, the symptoms of ToCV in tomatoes are atypical compared to those of other plant viruses. Symptoms tend to appear first in older leaves rather than younger ones and can often be confused with nutritional disorders, leading to its misidentification or delayed detection [29]. Finally, ToCV likely went undetected before the 1990s due to its low occurrence, which was a result of its transmission by whiteflies. The whitefly population only surged in the 1990s, enabling the virus to spread and be identified [30].

Our findings reveal that ToCV has disseminated across multiple regions, and this widespread diffusion appears to have shaped its genetic structure, which shows a weak geographic correlation. While the exact mechanisms driving such frequent dissemination remain unclear, long-distance migration events suggest that human activities, such as the movement of plant materials, trade, and travel, have played a central role. Notably, the most significant diffusion events occurred after the 1960s, coinciding with the rapid expansion of global trade, travel, and agricultural exchanges. This global interconnectedness likely facilitated the virus’s entry into new regions, enabling its spread beyond local boundaries.

However, it is important to note that long-distance dissemination primarily facilitated the spread of ToCV into new regions, rather than contributing significantly to its maintenance in these regions [Figure 2]. Once the virus became established in a region, intra-regional migration predominated from the 1960s onward. This shift suggests that local epidemics and viral persistence were primarily sustained through regional transmission dynamics, likely driven by vectors such as whiteflies and agricultural practices, including the movement of infected plants within local farming systems. These findings highlight the critical importance of both quarantine measures to prevent the introduction of the virus into new areas and local management strategies aimed at controlling its spread within established regions.

While our Bayesian phylogeographic analysis provides useful insights into the spread of ToCV, it was unable to precisely identify its origin. Several factors may have contributed to this limitation. First, the virus’s complex migration patterns, which include both long-distance and local movements, make it challenging to pinpoint a single origin. Second, the sequence data used in this study were primarily retrieved from publicly available databases, which resulted in uneven geographic representation. This likely led to gaps in regions where the virus circulated undetected, particularly in wild plant populations. Additionally, the virus’s subtle symptoms and broad host range could have allowed it to persist unnoticed in certain areas, complicating efforts to trace its emergence.

To further refine our understanding of ToCV’s origins, future studies might benefit from expanded geographic sampling, including more diverse reservoirs, and improved detection methods to uncover earlier, unnoticed spread.

## Figures and Tables

**Figure 1 viruses-17-00457-f001:**
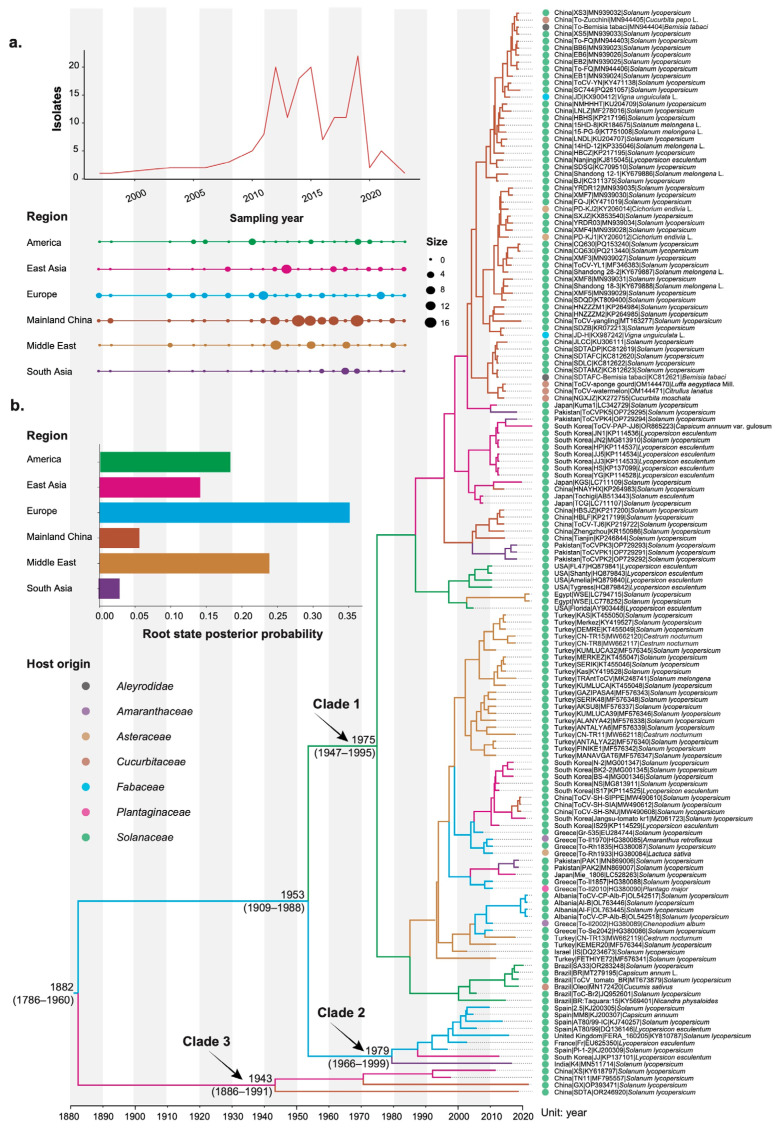
Overview of viral isolate sampling and a time-scaled tree of tomato chlorosis virus (ToCV). (**a**) The total sample sizes and the distribution of ToCV isolates across the geographic locations over time (year). Each region is denoted by a unique color, and dot sizes correspond to the relative sample sizes. (**b**) A maximum clade credibility tree inferred from CP sequences of 155 recombination-free isolates. The tree topology was selected to maximize the product of node posterior probabilities. Branches are color-coded to reflect their geographic origins, while colored circles at the tips represent the host origin, as defined in the accompanying color legend. The inset panel shows the inferred probabilities of the root location for each region.

**Figure 2 viruses-17-00457-f002:**
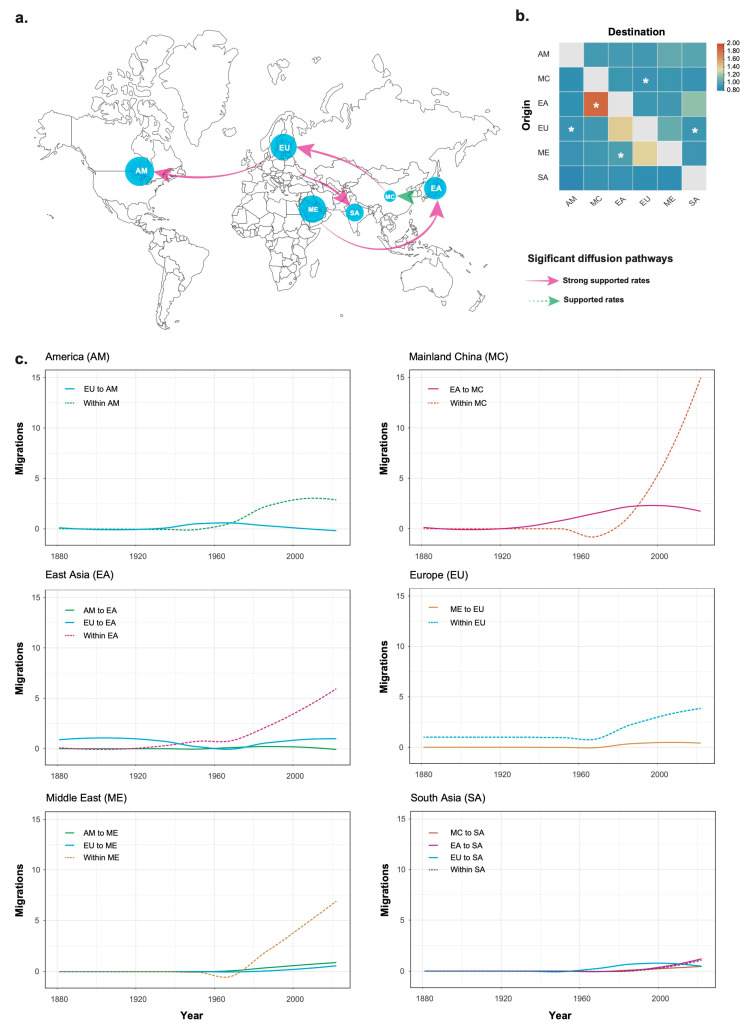
Phylogeographic reconstruction of tomato chlorosis virus (ToCV) spread. (**a**) Spatial diffusion pathways inferred from 155 ToCV isolates, all supported by 10 subsampled datasets. Line corresponds to migration rates with mean indicator of >0.5. Solid pink arrows denote strongly supported migration (Bayes factor, 20 < BF < 150), while dashed green arrows indicate moderately supported rates (3 < BF < 20). Dot sizes are proportional to median inferred effective population sizes. AM, America; MC, Mainland China; EA, East Asia; EU, Europe; ME, Middle East; SA, South Asia. (**b**) Heatmaps showing mean migration rate across geographic regions. Significant migration events, characterized by mean indicator of >0.5 and BF > 3, are marked with asterisks. (**c**) Migration events (on a log10 scale) of ToCV through time, including within-region migration for America, Mainland China, East Asia, Europe, Middle East, and South Asia.

## Data Availability

All data used in this study are publicly available on NCBI. A list of the accession numbers used is found in Appendix A.

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
