# Peer review of "Phylogeographic History of Tomato Chlorosis Virus"

_viruses, 2025, doi:10.3390/v17040457_

Round 1
Reviewer 1 Report
Comments and Suggestions for Authors
In the manuscript “Phylogeographic history of tomato chlorosis virus” by Wu and colleagues, the authors report the results of a phylogeographic analysis of ToCV, a member of the genus Crinivirus, posing serious threat to tomato cultivation worldwide. The authors conducted a Bayesian phylogeographic analysis based on the coat protein sequence of 155 ToCV isolates, retrieved from GenBank and in part determined by the same authors. Prior to the analyses the authors strictly considered the input sequence scenario, and verified the absence of recombinant sequences, which would affect the correct estimation of evolutionary rates, and additionally prepared a subset of datasets to conduct parallel analysis in order to avoid bias due to the high number of sequences from mainland China. Initial evaluation of the presence of a temporal structure in the data by different methods was inconclusive, therefore the substitution rate of the CP gene was used for estimation of the evolutionary timescale. The spatio-temporal dynamics were investigated in BEAST. Three distinct clades grouping the isolates were defined, and five significant migration pathways were identified. The likely geographic origin of ToCV could not be determined and authors provides justification for that in the Discussion. The analysis provides insights into the spread pathways of ToCV.
A parallel analysis with NextStrain, a platform widely used for tracking pathogen evolution, would allow to compare different approaches and the results obtained. I suggest the authors to consider performing this analysis as they have the sequence dataset ready for analysis. Furthermore, analysis on gene flow, genetic diversification and natural selection on the CP could be added to enrich the Communication.
Beside the above consideration, I invite the authors to consider the following minor comments and suggestions:
L23: regions =˃ geographic areals
L31: =˃ including member of ….
L33: cursive for botanic family name
L43: =˃ composed of two positive….
L47: =˃ essential for assemblying the….
L51: studies at the genome level may be also limited by the number of (nearly) complete genome sequences, therefore making gene sequences as the only source for the analyses. The authors could add on this in the text.
L54: =˃ of the genera…
L101: better specify the meaning of “previous inference” and the source
L226-229: the authors should add more on the basis (literature? previous works?) that support these considerations on the intraregional migration
Author Response
In the manuscript “Phylogeographic history of tomato chlorosis virus” by Wu and colleagues, the authors report the results of a phylogeographic analysis of ToCV, a member of the genus Crinivirus, posing serious threat to tomato cultivation worldwide. The authors conducted a Bayesian phylogeographic analysis based on the coat protein sequence of 155 ToCV isolates, retrieved from GenBank and in part determined by the same authors. Prior to the analyses the authors strictly considered the input sequence scenario, and verified the absence of recombinant sequences, which would affect the correct estimation of evolutionary rates, and additionally prepared a subset of datasets to conduct parallel analysis in order to avoid bias due to the high number of sequences from mainland China. Initial evaluation of the presence of a temporal structure in the data by different methods was inconclusive, therefore the substitution rate of the CP gene was used for estimation of the evolutionary timescale. The spatio-temporal dynamics were investigated in BEAST. Three distinct clades grouping the isolates were defined, and five significant migration pathways were identified. The likely geographic origin of ToCV could not be determined and authors provides justification for that in the Discussion. The analysis provides insights into the spread pathways of ToCV.
A parallel analysis with NextStrain, a platform widely used for tracking pathogen evolution, would allow to compare different approaches and the results obtained. I suggest the authors to consider performing this analysis as they have the sequence dataset ready for analysis. Furthermore, analysis on gene flow, genetic diversification and natural selection on the CP could be added to enrich the Communication.
RESPONSE: Thank you for your suggestion. We agree that comparing different approaches and the results obtained would be valuable. We certainly plan to conduct this analysis in the future. However, we are not currently familiar with NextStrain, so we will explore this platform and incorporate it into future studies. Regarding your suggestion to include analyses on gene flow, genetic diversification, and natural selection on the CP, we have indeed conducted some of these analyses. However, we found it challenging to incorporate them into this particular communication. Additionally, similar analyses have already been reported by other authors, although with slightly different datasets.
Beside the above consideration, I invite the authors to consider the following minor comments and suggestions:
L23: regions =˃ geographic areas
RESPONSE: Changed as suggested.
L31: =˃ including member of ….
RESPONSE: Revised as suggested.
L33: cursive for botanic family name
RESPONSE: Revised as suggested.
L43: =˃ composed of two positive ….
RESPONSE: Revised as suggested.
L47: =˃ essential for assembling the….
RESPONSE: Revised as suggested.
L51: studies at the genome level may be also limited by the number of (nearly) complete genome sequences, therefore making gene sequences as the only source for the analyses. The authors could add on this in the text.
RESPONSE: Thank you for your suggestion. We have added the following statement in the text to justify our focus on CP.
L54: =˃ of the genera…
RESPONSE: Revised as suggested.
L101: better specify the meaning of “previous inference” and the source.
RESPONSE: We have now rephrased the sentence, with citation of the relevant reference.
L226-229: the authors should add more on the basis (literature? previous works?) that support these considerations on the intraregional migration
RESPONSE: These statements are interpretations based on the data presented in Figure 2, which has been cited here.
Reviewer 2 Report
Comments and Suggestions for Authors
Wu and colleagues conducted a Bayesian phylogeographic analysis of CP gene sequences from 155 ToCV isolates to reconstruct its phylogeographic history. Their findings suggest that ToCV may have emerged around 1882 CE. They identified five migration pathways, three involving Europe and two involving East Asia. The manuscript provides valuable insights. However, i have a few minor comments
- In line 33, Solanaceae should be italic.
The author states that 155 sequences dated from 2003 to 2023 were used. However, in the dataset (Table S1), two sequences (KJ200309 and MF795557) from 1997 and 1998 were included. These sequences also appear in Figure 1. Therefore, the information on line 70, page 2, should be updated to reflect the correct date range as 1997–2023 instead of 2003–2023.
-In the Discussion section, the author stated: "The most recent common ancestor (MRCA) of ToCV is dated back to 1882, more than 100 years before its first detection in 1998. This implies that ToCV could have been circulating undetected for over a century before it was formally identified as a pathogen." I agree with this statement. Additionally, further supporting evidence is that while ToCV was formally identified in 1998, the yellow leaf disorder syndrome, a symptom associated with ToCV was observed as early as 1989 in Florida.
-The reference 29 cited in line 213 does not appear to be appropriate for the virus in question. After reading the paper, it does not specifically describe ToCV symptoms. Instead, a more suitable reference would be Wisler et al., 1998, which provides a more detailed description of ToCV symptoms: (Wisler, G.C. , Duffus, J.E. , Liu, H.‐Y. and Li, R.H. (1998) Ecology and epidemiology of whitefly‐transmitted closteroviruses. Plant Dis. 82, 270–280).
-The sentences in lines 213–216 are unclear and need to be revised for better clarity. Additionally, after checking reference 30, this cited paper does not suggest that there was a surge in whitefly populations in the late 1990s as you wrote.
Author Response
Wu and colleagues conducted a Bayesian phylogeographic analysis of CP gene sequences from 155 ToCV isolates to reconstruct its phylogeographic history. Their findings suggest that ToCV may have emerged around 1882 CE. They identified five migration pathways, three involving Europe and two involving East Asia. The manuscript provides valuable insights.
RESPONSE: We thank the reviewer for these positive comments.
However, i have a few minor comments
- In line 33, Solanaceae should be italic.
RESPONSE: Changed as suggested.
The author states that 155 sequences dated from 2003 to 2023 were used. However, in the dataset (Table S1), two sequences (KJ200309 and MF795557) from 1997 and 1998 were included. These sequences also appear in Figure 1. Therefore, the information on line 70, page 2, should be updated to reflect the correct date range as 1997–2023 instead of 2003–2023.
RESPONSE: We have now corrected the date range to reflect the accurate timeline of 1997–2023.
-In the Discussion section, the author stated: "The most recent common ancestor (MRCA) of ToCV is dated back to 1882, more than 100 years before its first detection in 1998. This implies that ToCV could have been circulating undetected for over a century before it was formally identified as a pathogen." I agree with this statement. Additionally, further supporting evidence is that while ToCV was formally identified in 1998, the yellow leaf disorder syndrome, a symptom associated with ToCV was observed as early as 1989 in Florida.
RESPONSE: Thank you for your feedback. Based on the information you provided, we have revised the sentence to: "The most recent common ancestor (MRCA) of ToCV is dated to 1882, more than 100 years before the yellow leaf disorder syndrome associated with this virus was first observed in Florida in 1989."
-The reference 29 cited in line 213 does not appear to be appropriate for the virus in question. After reading the paper, it does not specifically describe ToCV symptoms. Instead, a more suitable reference would be Wisler et al., 1998, which provides a more detailed description of ToCV symptoms: (Wisler, G.C., Duffus, J.E. , Liu, H.‐Y. and Li, R.H. (1998) Ecology and epidemiology of whitefly‐transmitted closteroviruses. Plant Dis. 82, 270–280).
RESPONSE: We thank the reviewer for pointing this out. We have replaced the inappropriate reference with the suggested citation, Wisler et al., 1998, which provides a more detailed description of ToCV symptoms. The correction has been made in the manuscript.
-The sentences in lines 213–216 are unclear and need to be revised for better clarity. Additionally, after checking reference 30, this cited paper does not suggest that there was a surge in whitefly populations in the late 1990s as you wrote.
RESPONSE: We thank the reviewer for their valuable feedback. In the revised manuscript, the sentences in lines 213–216 have been revised as follows:
"Finally, ToCV likely went undetected before the 1990s due to its low occurrence, which resulted from its transmission by whiteflies. Whitefly populations began to surge in the 1990s, facilitating the virus’s spread and eventual identification." Additionally, the citation has been corrected as follows: [Navas-Castillo, J., Fiallo-Olivé, E., & Sánchez-Campos, S. (2011). Emerging virus diseases transmitted by whiteflies. Annual review of phytopathology, 49(1), 219-248.]
Round 2
Reviewer 1 Report
Comments and Suggestions for Authors
The authors have replied to the minor comments and edited the manuscript. It would have been of interest to add the data of the genetic analyses on the CP.
Comments on the Quality of English Languageok